# Experimental Study of Energy Evolution at a Discontinuity in Rock under Cyclic Loading and Unloading

**DOI:** 10.3390/ma15165784

**Published:** 2022-08-22

**Authors:** Wei Zheng, Linlin Gu, Zhen Wang, Junnan Ma, Hujun Li, Hang Zhou

**Affiliations:** 1Department of Civil Engineering, Nanjing University of Science and Technology, Nanjing 210094, China; 2School of Mechanical Engineering, Nanjing University of Science and Technology, Nanjing 210094, China; 3Department of Civil Engineering, Nagoya Institute of Technology, Nagoya 466-8555, Japan

**Keywords:** rock mechanics, rock discontinuity, energy evolution, energy dissipation ratio

## Abstract

Energy is often dissipated and released in the process of rock deformation and failure. To study the energy evolution of rock discontinuities under cyclic loading and unloading, cement mortar was used as rock material and a CSS-1950 rock biaxial rheological testing machine was used to conduct graded cyclic loading and unloading tests on Barton’s standard profile line discontinuities with different joint roughness coefficients (JRCs). According to the deformation characteristics of the rock discontinuity sample, the change of internal energy is calculated and analyzed. The experimental results show that under the same cyclic stress, the samples harden with the increase in the number of cycles. With the increase of cyclic stress, the dissipated energy density of each stage gradually exceeds the elastic energy density and occupies a dominant position and increases rapidly as failure becomes imminent. In the process of increasing the shear stress step-by-step, the elastic energy ratio shows a downward trend, but the dissipated energy is contrary to it. The energy dissipation ratio can be used to characterize the internal damage of the sample under load. In the initial stage of fractional loading, the sample is in the extrusion compaction stage, and the energy dissipation ratio remains quasi-constant; then the fracture develops steadily, the damage inside the sample intensifies, and the energy dissipation ratio increases linearly (albeit at a low rate). When the energy storage limit is reached, the growth rate of energy dissipation ratio increases and changes when the stress level reaches a certain threshold. The increase of the roughness of rock discontinuity samples will improve their energy storage capacity to a certain extent.

## 1. Introduction

There are many different types of rock discontinuities in natural rock mass, such as joints, folds, and faults. The existence of these rock discontinuities causes discontinuity and anisotropy in mechanical properties of rock mass, which directly affects the stability of rock mass [1,2,3]. In underground caverns, dams, abutments, and other rock mass projects, cyclic loads are common such as those induced by blasting, earthquakes, and changes in water level. Under the long-term action of these cyclic loads, rock mass discontinuities may slip and deform, resulting in failure of the rock mass structure [4,5]. Invoking the second law of thermodynamics, energy conversion is a basic feature of material physical processes, the essence of material damage is energy to drive the state of instability [6], and the failure of rock is an irreversible process of energy dissipation [7,8]. Therefore, studying the damage and failure of rock discontinuity from an energy perspective can provide a new idea for preventing and treating rock engineering disasters.

In recent years, scholars have conducted a series of studies on the mechanisms underpinning the evolution of the internal energy of a rock mass. For example, Xie et al. [9,10] proposed that failure is due to the abrupt change of energy dissipation under certain conditions and defined a rock strength failure criterion. In combination with damage mechanics and energy conservation theory, Tu et al. [11] established a slope instability criterion using a strength-reduction method based on energy conversion. Peng et al. [12] studied the relationship between crack angle and energy inside the loaded rock, and the results showed that angle was positively correlated with energy storage capacity. Wang et al. [13] used numerical simulation software to calculate the energy storage of surrounding rock, expounded the relationship between rock failure and elastic energy, and confirmed that the simulated strain energy analysis method could be used for rockburst prediction. Wu et al. [14] studied the energy evolution of rock under different loading modes. Wang et al. [15] conducted a uniaxial cyclic charge test on the lower dry and saturated sandstone, and analyzed the strength and deformation characteristics of the rock, as well as the change and distribution of energy under the dry and saturated state. Jia et al. [16] studied the energy variation law of rock mass at different depths during mining, and the results showed that the increase of the depth of rock mass would lead to the increase of all types of energy inside it. Deng et al. [17] carried out dynamic uniaxial compression tests on rock samples under impact velocity and analyzed the energy dissipation law in the dynamic failure process of rock. Zhang et al. [18] studied the influence of confining pressure on the change of energy inside the rock in the triaxial test, and the results showed that the increase of confining pressure could improve the efficiency of energy accumulation to a certain extent. Munoz et al. [19,20] defined a new rock brittleness index to elucidate the energy accumulation and release of rock failure. Song et al. [21] performed uniaxial cyclic loading and unloading tests on coal and rock, collected electromagnetic radiation signals released in the test process, and established the correlation between electromagnetic radiation and dissipated energy.

The above research has greatly enriched the application of energy theory in rocks, but most scholars focus on intact rocks and have little research on rock discontinuity, and in the cyclic loading and unloading test, often with unloading in the next cycle to 0 MPa, by changing the upper limit of cyclic loading for different stress amplitudes. The lower limit of cyclic load is often not zero, and therefore in the present work cement mortar was used to represent the rock, and a CSS-1950 biaxial rheological testing machine was adopted to conduct graded cyclic loading and unloading tests on Barton’s standard profile line discontinuities of different joint roughness coefficients (JRCs). The energy evolution of samples in the process of failure was revealed, which can provide a theoretical basis for studying rock damage and failure mechanisms from the perspective of energy.

## 2. Test Equipment and Specimens

### 2.1. Test Equipment

The CSS-1950 (Model of testing machine, CSS is creep shear strength) rock biaxial rheological testing machine was used in this test (Figure 1a), which is manufactured by the Changchun Institute of Testing Machines in Changchun, China. The test machine includes vertical and horizontal loading systems with maximum vertical and horizontal compression loads of 500 and 300 kN, respectively. Two linear variable displacement transducers (LVDT) were used with a measurement range of 0–10 mm and accuracy of 0.0001 mm to monitor vertical and horizontal deformation, as shown in Figure 1b. The test machine adopts servo-motor control, a pressure system for screw pressure, load-rate control, and a continuous working time of more than 1000 h.

### 2.2. Sample Preparation

Due to the random composition and surface morphology of natural rock mass, it is difficult to prepare relatively uniform samples using natural rock mass, and it is impossible to quantify the roughness of the rock discontinuity, which results in difficulty when comparing test results; therefore, because cement mortar has good uniformity, it is often used as a kind of rock material to simulate rock discontinuities [22,23]. Therefore, in the present work, cement mortar was used as a similar material to prepare Barton’s standard profile line discontinuities with different JRCs for testing.

In this test, the cement mortar samples named 1^#^, 4^#^, 8^#^, and 10^#^ were respectively adopted to represent the features of structural planes with different roughness. Their Barton’s standard profile line features are shown in Figure 2b. For the convenience of analysis, the JRC of the sample was taken as the intermediate value, that is, the cement mortar samples named 1^#^, 4^#^, 8^#^, and 10^#^ were, respectively, 1, 7, 15, and 19 herein.

The steel mold used in this test was made by referring to Barton’s standard profiles and using high precision (0.1 μm) computer control [24], as shown in Figure 2a. According to need, the upper and lower parts of the sample were prepared, and the two parts were combined to form a complete rock discontinuity sample (Figure 2c). Each specimen measured 100 × 100 × 100 mm.

The sample material was Portland cement with a compressive strength of 32.5 MPa, standard sand and water, and the mixing ratio was sand/cement/water of 4:2:1. After mixing evenly, the mold was filled. After filling, the sample was removed after the cement mortar was formed. After removal, the samples were placed in the curing chamber and stored in the standard curing chamber at the temperature and humidity of (20 ± 1) °C and 95%, respectively, for 28 days before testing.

According to the test requirements, a total of 29 specimens were prepared, including 5 intact samples and 24 rock discontinuity samples (the cement mortar samples named 1^#^, 4^#^, 8^#^, and 10^#^ were prepared in 6 pieces each). In total, the intact samples were used for the uniaxial compression test, and the rock discontinuity samples were used for the direct shear test and graded cyclic shear test.

## 3. Test Procedure

### 3.1. Uniaxial Compression Test

To determine the normal stress value of shear test, uniaxial compression tests were conducted on five complete cement mortar specimens at a loading rate of 0.4 kN/s. The test results are shown in Table 1. Here, 10%, 20%, and 30% of the average compressive strengths were taken as the normal stresses of subsequent shear tests, which were 2.17, 4.35, and 6.52 MPa, respectively.

### 3.2. Direct Shear Test

To obtain the shear strength of rock discontinuities, the cement mortar samples named 1^#^, 4^#^, 8^#^, and 10^#^ (JRC = 1, 7, 15, 19) were selected and shear tests were conducted at a rate of 0.2 kN/s under the normal stress of 2.17, 4.35, and 6.52 MPa, respectively, until failure. The shear strength obtained was used as the basis for the classification of shear load grades in subsequent cyclic shear tests, and Barton’s standard profile line number and normal stress were used to number the test results. For example, the test results of rock discontinuity No. 1 under the normal stress of 2.17 MPa were denoted “1–2.17”, and the test data are listed in Table 2.

### 3.3. Graded Cyclic Shear Tests

The cement mortar samples named 1^#^, 4^#^, 8^#^, and 10^#^ (JRC = 1, 7, 15, 19) were selected to conduct cyclic shear tests under normal stresses of 2.17, 4.35, and 6.52 MPa, respectively. The loading of samples is shown in Figure 3a. The normal stress was first added to a predetermined value, and the shear stress was applied after the normal deformation was stabilized. The shear stress was divided into multiple stages. The upper limit of the shear stress at the first stage was 30% of the shear strength, and each stage increased the stress by 10% of the shear strength. Under the same level of loading, the cyclic amplitude of shear stress was 10% of the shear strength. Under the same level of loading, 10 cycles were carried out with a loading rate of 0.2 kN/s. Continuous loading was conducted until failure. The actual loading stress is listed in Table 3.

## 4. Results and Discussion

### 4.1. Deformation Characteristics of Specimen during Failure

The purpose of this research was to reveal the energy evolution of rock discontinuity under cyclic loading. Due to the limitation of word count, only one group of test data was studied, and the stress–strain relationship of other samples is similar, so it will be repeated here.

Figure 4 displays the whole process of the stress and displacement curve for sample 4–6.52. As shown in the figure, during each cycle, the unloading curve does not coincide with the original loading curve, and the unloading curve can form a completely closed annular area with the reloading curve, namely, the hysteresis loop [25]. The reason is that rock materials are not ideal elastomers, and there are a large number of internal defects such as pores and micro-cracks, which lead to the closure of the original cracks and the initiation of new cracks in the test process [26], resulting in some irreversible damage to the sample. The appearance of the hysteresis loop is not only an experimental phenomenon, but also a manifestation of energy dissipation, and the area of the hysteresis loop can be characterized as the energy dissipated by crack closure, expansion, and through-cracking failure of the loaded sample; the larger the area of the hysteresis loop, the more energy consumed and the more severe the damage to the sample.

The rock discontinuity sample is different from the intact rock sample, and in the first loading process of each grading cycle stage, the upper and lower parts of the rock discontinuity sample usually slip, resulting in large residual deformation. With the continuous increase of stress, this sliding deformation gradually increases, and the deformation curve of the sample is gradually sparse, indicating that the internal damage of the sample is increasingly aggravated with the increase of stress, and its ability to resist shear is gradually decreased, resulting in the gradual increase of irreversible deformation. However, under the same cyclic stress, the stress–strain curve changes from thinning to dense, and the hysteresis loop area decreases gradually, indicating that the specimen gradually compacts with the increase of cyclic number and the hardening degree increases.

### 4.2. Energy Density Calculation Method

For the loaded sample, it often goes through the extrusion compaction stage, the stable development stage, and the unstable development stage of the crack before the final failure, and these processes are often accompanied by energy input, accumulation, dissipation, and release. The energy transformation of samples from deformation to failure is a dynamic process, which is represented by the transformation and balance among input energy, elastic energy, and dissipated energy. According to the first law of thermodynamics, the energy of substances in a thermodynamic system can be transformed and transferred, and the total amount of energy remains unchanged in the process of transformation and transfer [6]. The work performed by the surrounding to the rock mass can produce energy input into the rock mass, causing reversible deformation and irreversible deformation of the rock mass. The reversible deformation accumulates in the form of elastic energy. Irreversible deformation dissipates energy in the form of plastic deformation energy, internal friction of rock mass, and thermal radiation [27]. The sample in the test is deformed by external force loading and is a closed system. Therefore, the energy exchange between the specimen and the outside world is not considered [28,29]. According to the law of the conservation of energy,
(1)$$U=U_{e}+U_{d}$$
where *U* is the total energy input from the outside; *U*_e_ denotes the elastic energy; and *U*_d_ is the dissipated energy, that is, the sum of the damage energy and plastic strain energy in rock mass.

At any time in the deformation process of the sample, there is a specific energy state corresponding to it [30], which is a function of stress, strain, and time. According to the stress–strain curve characteristics of samples, the elastic energy density and dissipation energy density of rock mass under cyclic loading were calculated. The relationship between the elastic energy density *u*_e_ and dissipation energy density *u*_d_ per unit of volume in the loading and unloading curve of samples under certain stress levels is shown in Figure 5. The total energy density *u* input by the outside world is the area formed by *OBAE*. The elastic energy density *u*_e_ denotes the area formed by *ACDE*. The dissipative energy density *u*_d_ is the total energy density *u* minus the elastic energy density *u*_e_, which is the area formed by *OBACD*. The *u*_e_ and *u*_d_ are calculated as follows:(2)$$u_{e}=\int_{\varepsilon_{1}}^{\varepsilon_{2}}\sigma d\varepsilon$$
(3)$$u_{d}=\int_{\varepsilon_{0}}^{\varepsilon_{2}}\sigma d\varepsilon -\int_{\varepsilon_{1}}^{\varepsilon_{2}}\sigma d\varepsilon$$
where *σ* is the stress at any point in the stress–strain curve; *ε* denotes the strain corresponding to *σ*; *ε*_0_ refers to the strain corresponding to the initial stress *σ*_0_ of the loading curve; *ε*_1_ is the strain corresponding to the lower limit stress *σ*_0_ of the unloading curve; and the strain corresponding to the loading of the upper limit stress *σ*_1_ is represented by *ε*_2_.

### 4.3. Energy Evolution Process in Samples

According to the above, during the first loading of each stage of the graded cycle, rock discontinuities usually produce large residual deformation, which makes it difficult for the unloading curve to form a hysteresis loop with the reloading curve, and the error in the calculation of the energy density is large; therefore, the second cycle was explored as if it were the first cycle. According to Equations (2) and (3), the energy density of the sample in each cycle of loading and unloading can be calculated. Taking specimen 4–6.52 as an example, the relationship between the energy density and cycle times under various cyclic stresses is shown in Figure 6.

As illustrated in Figure 6, the external input energy shows an overall downward trend with the increase in the number of cycles, but its change process is different under different cyclic stresses. When the stress is small (1.35 to 2.25 MPa), the external input energy decreases gradually with increasing number of cycles, and the decrease is significant in the first few cycles, small in the later stage, then tends to be stable. When the stress is large (2.70 to 4.95 MPa), the external input energy decreases rapidly with the increase in the number of cycles, and then decreases again after a small increase.

With the increase in the number of cycles, the elastic energy density shows an overall upward trend, and increases significantly in the early stage, and gradually stabilizes in the later stage, while the dissipated energy density gradually decreases, which is consistent with the phenomenon that the area of the hysteresis loop in the stress–strain curve decreases with increasing number of cycles. The results show that in the cyclic loading and unloading process associated with the same stress level, the micro-cracks inside the sample gradually close, the hardening of the sample increases, and the energy consumed by the friction inside the sample decreases.

With the continuous increase of cyclic stress, the dissipated energy density of each stage gradually exceeds the elastic energy density and gradually occupies a dominant position, indicating that the internal damage to the sample is gradually intensified.

The energy evolution of the other samples at each stress is akin to that of specimen 4–6.52, which is not repeated in this paper due to limited space.

According to the actual failure strength of the sample, the cyclic stress is normalized, and the energy density of the last nine cycles under all levels of cyclic stress is averaged to obtain the relationship between the average energy density and the stress level (the ratio of the upper limit of cyclic stress at all levels to the actual failure strength of the sample, as shown in Table 4), as shown in Figure 7. With the increase in stress, the energy absorbed by the sample from the outside firstly decreases greatly, and then decreases significantly. When the stress reaches a certain value, it shows a significant upward trend. The elastic energy density presents a non-linear decreasing trend with the increase of stress, and the rate of change decreases gradually thereafter. With the increase in stress, the dissipated energy density decreases slightly at first, then increases slowly at a low rate, and increases significantly as the specimen approaches failure.

At the initial stage of fractional loading, the three energies all decreased. This phenomenon is due to the existence of many pores and cracks in the sample itself. In the process of first-stage cyclic loading and unloading, the closure and friction between these pores and cracks need to absorb more energy, and the absorbed energy is mainly stored in the sample in the form of elastic energy. Thus, the energy of each part of the sample is high in the first stage, and after entering the next stage of the cycle, most of the pores inside the sample have closed, and the degree of compaction is significantly improved, resulting in a decrease in the energy of all parts. With the further increase in stress, the primary cracks in the sample begin to expand, and new cracks are constantly initiated, leading to the decline of the ability of the sample to accumulate elastic energy. The energy consumed by internal friction and plastic failure increases gradually, which implies that the elastic energy density decreases continuously, while the dissipated energy density increases gradually.

### 4.4. Energy Distribution in Samples

In the closed test system, the energy input to rock samples by the testing machine is mainly transformed into elastic energy and dissipated energy, which will affect the deformation and failure of rock samples. The relationship between the proportion division of types of energy in samples and stress level is shown in Figure 8. With the constant increase in stress, the proportion of elastic energy and the proportion of dissipated energy in the sample show a non-linear trend, and the elastic energy presents a downward trend as a whole, while the dissipated energy shows the opposite trend, and the rates of change of both gradually increase.

At the initial stage of fractional loading, the internal pores of the sample are compressed and compacted, and the proportion of elastic energy is much higher than that of dissipated energy, indicating that most of the energy input from the outside is converted into elastic energy and stored in the sample. The energy consumed by internal crack closure and friction slip is small. With the increase in stress, the elastic energy ratio decreases step-by-step at a low rate, while the dissipated energy ratio changes in the opposite way. At this stage, although most of the pores inside the sample are closed, the stress concentration leads to the expansion and initiation of micro-cracks, increasing the dissipated energy ratio, but most of the energy from the external input is still accumulated in the sample. When the ratio of dissipated energy and elastic energy approach each other, the connection of micro-cracks in the sample and the formation and unstable expansion of macro-cracks lead to the dissipation of most of the external input energy. The elastic energy accumulated in the sample begins to release, showing that the proportion of dissipated energy decreases rapidly.

### 4.5. Energy Criterion for Rock Discontinuity Failure

For an ideal material, any form of energy applied to it can all be converted into releasable elastic strain energy within the material, and during the deformation process, its internal structure does not suffer damage, and the absorption and storage of elastic energy will not dissipate. However, for rock materials, the friction of pores or micro-cracks inside the specimen in the compaction stage, and the expansion of micro-cracks and connection in the elastic and plastic stages are all accompanied by energy dissipation. The occurrence and accumulation of irreversible deformation are the direct causes of specimen failure [31], and the dissipative energy can indirectly reflect the irreversible deformation generated in the specimen. The accumulation of dissipative energy will facilitate the sample in its gradual transformation from the initial stable state to an unstable state, and then to another stable state (the strength therein being the residual strength after the main fractures have inter-connected and split into multiple rock blocks) through the reorganization of the internal structure of the sample. The change from a stable state to an unstable state entails the process of internal damage accumulation to unstable failure, as well as the process of internal energy transformation. Therefore, the energy dissipation ratio *K* (the ratio of the dissipated energy density to the elastic energy density of the loaded rock sample) is used to characterize the deterioration of the sample, and also indirectly reflect the state of the sample. *K* is given by
(4)$$K=\frac{u_{d}}{u_{e}}$$

When *K* < 1, it can be considered that the internal structure of the sample is in a relatively stable state, and the damage to it is small; when *K* = 1, it can be considered that the loaded rock sample reaches its energy storage limit, is in a critical state, and is about to enter the unstable development stage; when *K* > 1, it can be considered that the rock sample is in an unstable state.

Figure 9 shows the stress level–energy dissipation ratio deformation of samples under different working conditions and Figure 10 demonstrates the energy consumption ratio of each sample that varies with the stress level.

According to the sample energy dissipation ratio and shear deformation seen in Figure 9, the failure process of the sample can be roughly divided into three stages. In the early stage of fractional loading, the sample is in the extrusion compaction stage (the first stage), and the internal pores and cracks of the sample are closed in this stage. The hardening of the sample is significantly improved, resulting in the irreversible deformation of the rock discontinuity which decreases significantly, and the energy dissipation ratio remains quasi-constant. Then the sample enters the stage of steady crack propagation (the second stage), and the internal damage intensifies, and the dissipated energy increases continuously due to the increase of internal friction and plastic deformation. The sample gradually reaches its energy storage limit (*K* = 1), which shows that the deformation of the structural plane and energy dissipation ratio increase linearly with the increasing stress. With the further increase of the stress level, the sample enters the unstable crack development stage (the third stage). The connection and penetration of cracks results in the worsening of the plastic failure of the sample, the rapid increase of dissipative energy, and the energy accumulated in the sample begins to release. When the failure is near, both the deformation and energy consumption ratio of the rock discontinuity change dramatically, indicating that a large amount of plastic failure occurs in the sample. The stored energy in the sample is released instantly, leading to the rapid loss of bearing capacity and instability failure.

Figure 10 illustrates that the energy dissipation ratio of all samples has roughly the same variation with stress, showing slow growth in the early stage, and a significant increase in the growth rate after reaching the energy storage limit (*K* = 1). However, there are significant differences in the stress on all samples when reaching the energy storage limit.

Instability and failure of specimens occurred after reaching the energy storage limit and are the root cause of a rapid release of elastic energy. Energy storage limits of the samples at the corresponding stress level can describe the samples’ accumulated elastic energy capacity: the higher the stress level, the more the sample storage limits the strength or stiffness, and instability and failure are less likely.

To explore the influence of roughness on the energy storage capacity of a specimen, the stress corresponding to each sample at *K* = 1 in Figure 10 was taken as the ultimate stress associated with its energy storage, and the results under the action of three different normal stresses under the same roughness were averaged to determine the variations in the ultimate stress of the energy storage of the sample with the roughness (Figure 11). The energy storage limiting stress for the specimen shows a positive correlation with the increase of JRC; that is, the rougher the rock discontinuity, the greater its energy storage limit and stiffness. According to the linear fitting results in the figure, the energy storage capacity of the rock discontinuity sample increases by about 0.75% when the value of JRC increases by 1.

## 5. Conclusions

The following conclusions can be drawn following hierarchical cyclic loading and unloading tests and energy calculation and analysis of specimens containing discontinuities:(1)Compared with the stress–strain curve, the energy density can clearly reflect the internal deterioration of the rock discontinuity, so as to predict the failure of the rock discontinuity more accurately.(2)Under the same cyclic stress, the specimen gradually hardens with the increase in the number of cycles. With the increase of cyclic stress, the dissipated energy density of each stage gradually exceeds the elastic energy density and occupies a dominant position and increases rapidly as failure becomes imminent.(3)With the increase of stress level, the elastic energy proportion of the sample presents a downward trend, with a slow rate in the early stage, but decreases significantly as the sample approaches failure; the variation in the proportion of energy dissipated shows the opposite trend.(4)The energy dissipation ratio can be used to characterize internal damage to the sample under load. In the initial stage of loading, the sample is in the extrusion and compaction stage, and the energy dissipation ratio remains unchanged. Then, the fracture develops steadily, the damage in the sample intensifies, and the energy dissipation ratio increases linearly (albeit at a low rate). Before the specimen is about to fail, the change rate is accelerated, and then a sudden change occurs, indicating that the rapid release of energy is the fundamental reason for the failure of the rock discontinuity.(5)The increase of the roughness of rock discontinuity samples will improve their energy storage capacity to a certain extent: the higher the JRC of the rock discontinuity, the greater the energy storage limit and stiffness of the specimen.

## Figures and Tables

**Figure 1 materials-15-05784-f001:**
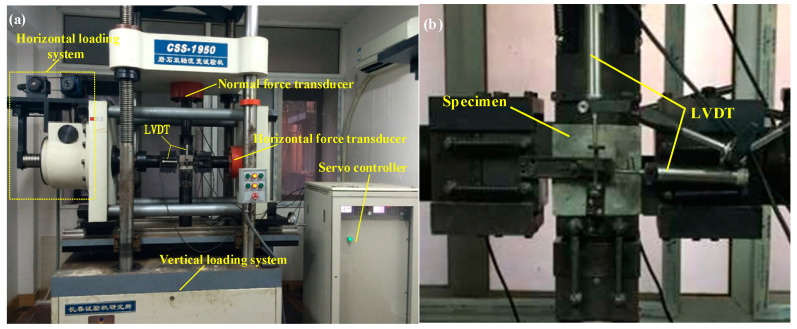
(**a**) CSS-1950 rock biaxial rheological testing machine; (**b**) specimen and monitoring system.

**Figure 2 materials-15-05784-f002:**
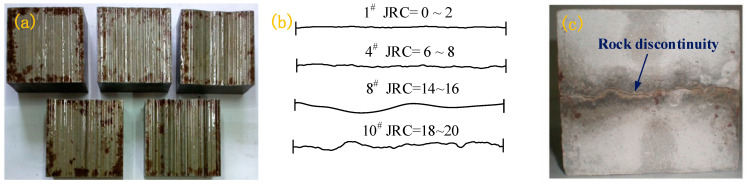
(**a**) Steel mold; (**b**) Barton’s standard profile lines; (**c**) sample.

**Figure 3 materials-15-05784-f003:**
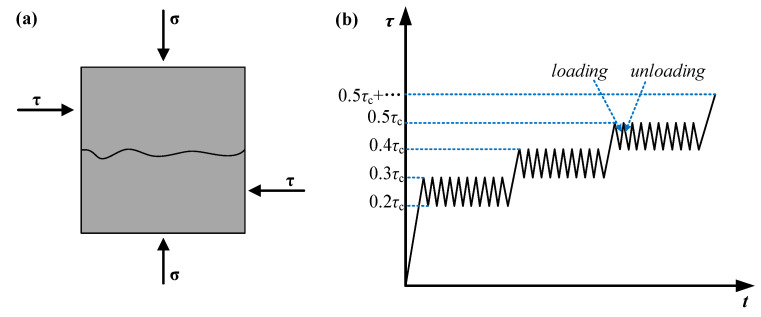
(**a**) Sample loading diagram; (**b**) schematic diagram of test loading path (*σ* is the normal stress and *τ* is the shear stress).

**Figure 4 materials-15-05784-f004:**
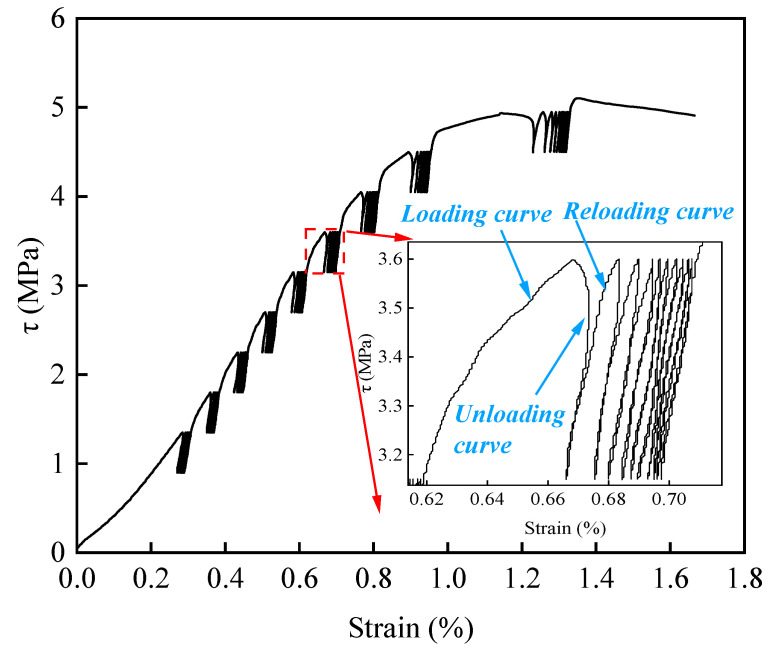
Stress–strain curves of specimen 4–6.52.

**Figure 5 materials-15-05784-f005:**
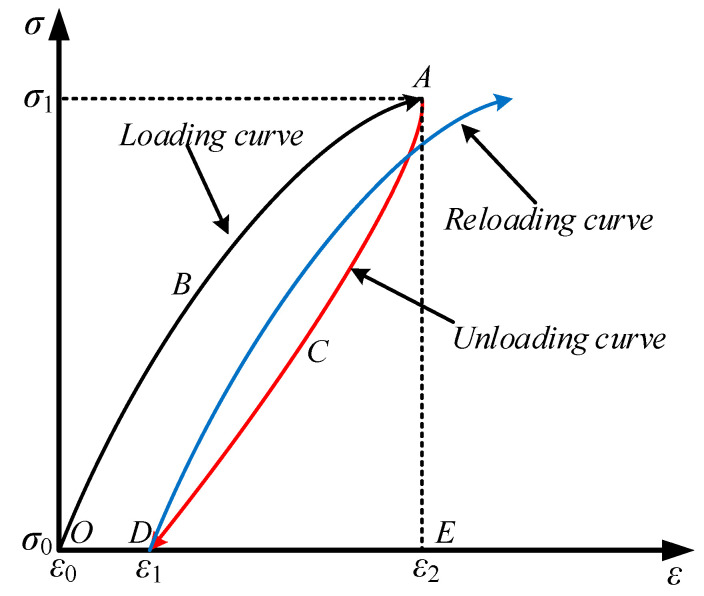
Schematic diagram of energy density calculation.

**Figure 6 materials-15-05784-f006:**
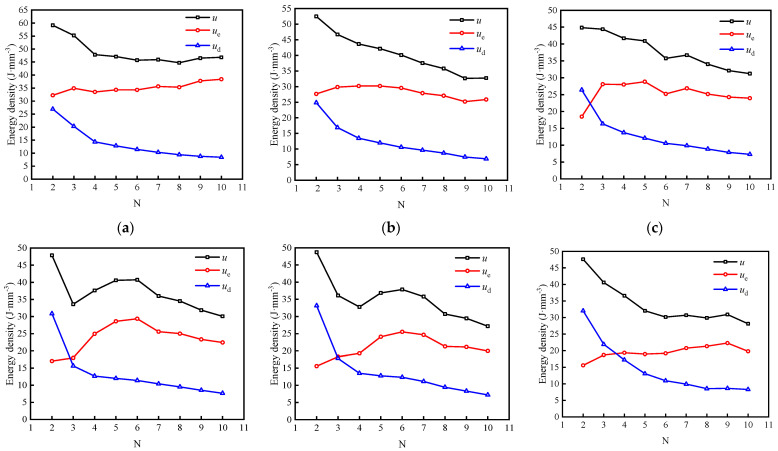
Variations in energy density of specimen 4–6.52 with the number of cycles *N* under various cyclic stresses: (**a**) 1.35 MPa; (**b**) 1.80 MPa; (**c**) 2.25 MPa; (**d**) 2.70 MPa; (**e**) 3.15 MPa; (**f**) 3.60 MPa; (**g**) 4.05 MPa; (**h**) 4.50 MPa; (**i**) 4.95 MPa.

**Figure 7 materials-15-05784-f007:**
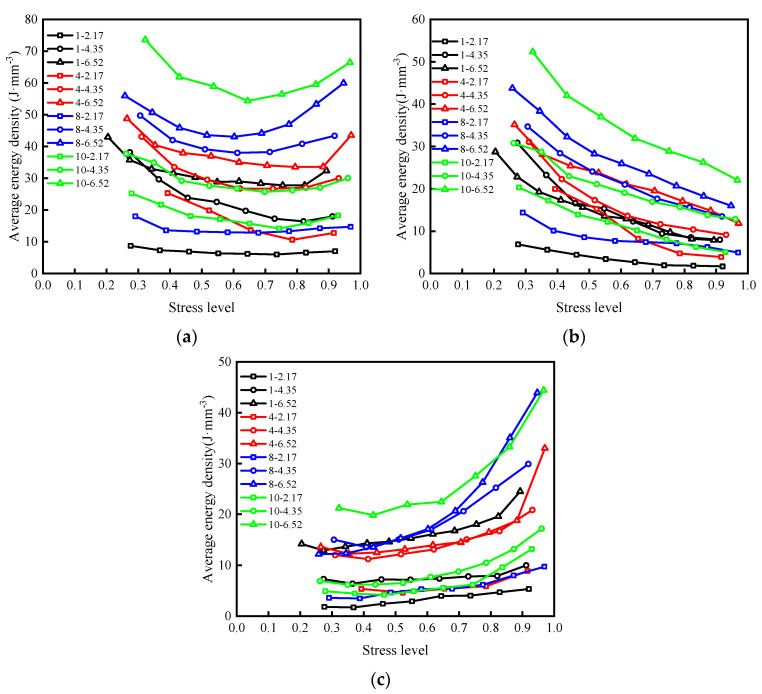
Relationship between energy density and stress level of samples under different working conditions: (**a**) *u*; (**b**) *u*_e_; (**c**) *u*_d_.

**Figure 8 materials-15-05784-f008:**
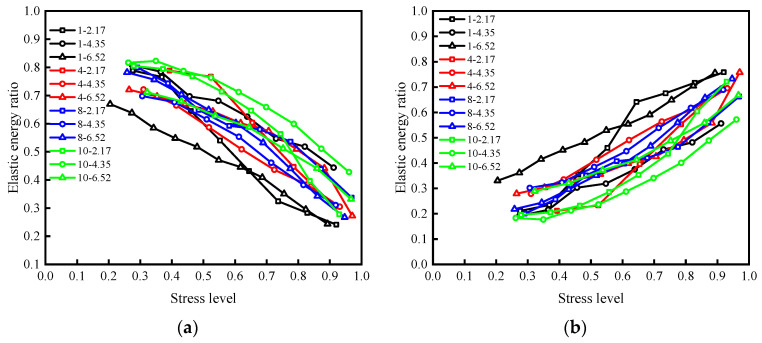
Relationship between energy ratio and stress level under different working conditions: (**a**) *u*_e_; (**b**) *u*_d_.

**Figure 9 materials-15-05784-f009:**
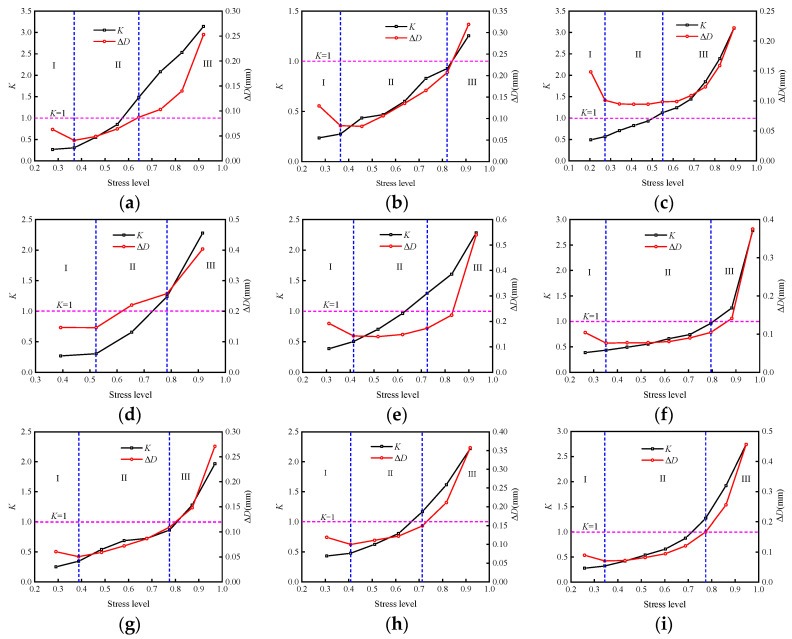
Stress level–energy dissipation ratio deformation of samples under different working conditions (Δ*D* is the total deformation of 10 cycles under different stress levels): (**a**) 1–2.17; (**b**) 1–4.35; (**c**) 1–6.52; (**d**) 4–2.17; (**e**) 4–4.35; (**f**) 4–6.52; (**g**) 8–2.17; (**h**) 8–4.35; (**i**) 8–6.52; (**j**) 10–2.17; (**k**) 10–4.35; (**l**) 10–6.52.

**Figure 10 materials-15-05784-f010:**
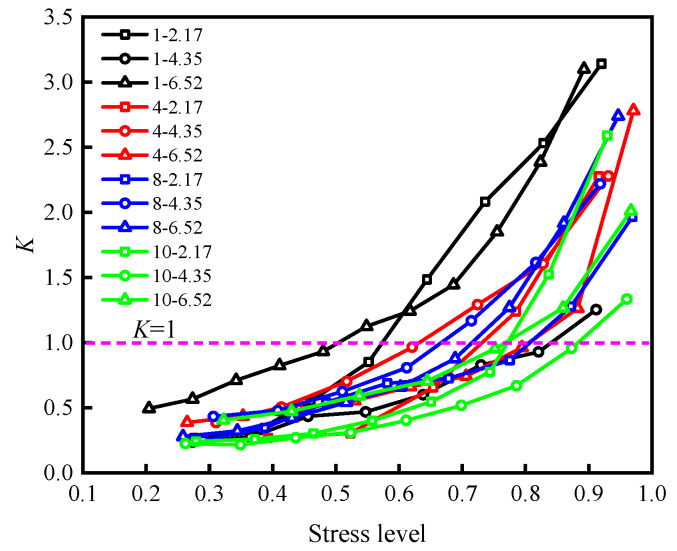
The energy consumption ratio of each sample varies with the stress level.

**Figure 11 materials-15-05784-f011:**
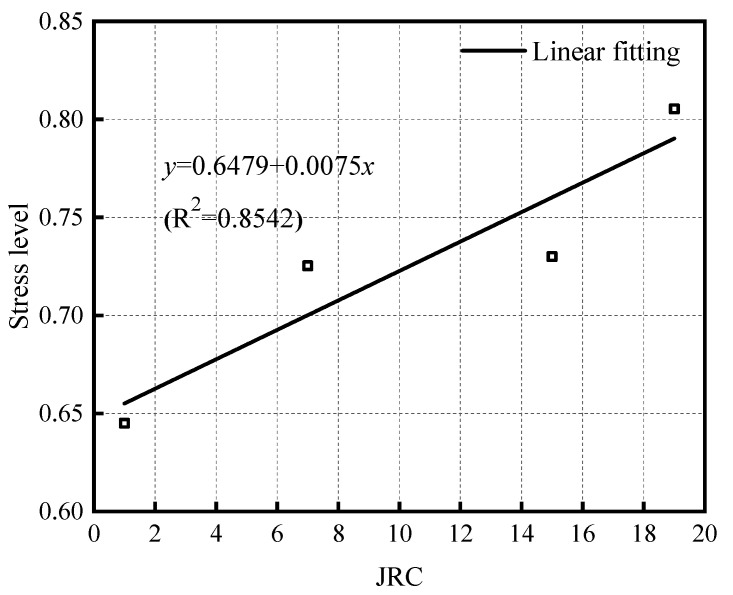
Variations in stress at the energy storage limit of samples with JRC.

**Table 1 materials-15-05784-t001:** Uniaxial compressive strength of each sample (*σ*_c_ is compressive strength).

Sample	Peak Stress/kN	*σ*_c_/MPa
1	228.47	22.85
2	233.44	23.34
3	177.69	17.77
4	248.37	24.84
5	198.55	19.86
Average value	217.30	21.73

**Table 2 materials-15-05784-t002:** Shear strength of each sample (*τ*_c_ is shear strength).

Sample	Peak Stress/kN	*τ*_c_/MPa
1–2.17	7.5	1.5
1–4.35	15.2	3.0
1–6.52	18.5	3.7
4–2.17	10.1	2.0
4–4.35	15.2	3.0
4–6.52	22.5	4.5
8–2.17	12.2	2.4
8–4.35	20.0	4.0
8–6.52	25.0	5.0
10–2.17	12.5	2.5
10–4.35	22.1	4.4
10–6.52	27.5	5.5

**Table 3 materials-15-05784-t003:** Loading stress table.

Sample	Upper Limit of Loading Stress at First Stage/MPa	Amplitude/MPa
1–2.17	0.45	0.15
1–4.35	0.90	0.30
1–6.52	1.10	0.37
4–2.17	0.60	0.20
4–4.35	0.90	0.30
4–6.52	1.35	0.45
8–2.17	0.72	0.24
8–4.35	1.20	0.40
8–6.52	1.50	0.50
10–2.17	0.75	0.25
10–4.35	1.32	0.44
10–6.52	1.65	0.55

**Table 4 materials-15-05784-t004:** Stress level.

Sample	Stress Level
1–2.17	0.28, 0.37, 0.46, 0.55, 0.64, 0.74, 0.83, 0.92
1–4.35	0.27, 0.36, 0.46, 0.55, 0.64, 0.73, 0.82, 0.91
1–6.52	0.20, 0.27, 0.34, 0.41, 0.48, 0.55, 0.62, 0.69, 0.75, 0.82, 0.89
4–2.17	0.39, 0.52, 0.65, 0.79, 0.92
4–4.35	0.31, 0.41, 0.52, 0.62, 0.72, 0.83, 0.93
4–6.52	0.26, 0.35, 0.44, 0.53, 0.62, 0.71, 0.79, 0.88, 0.97
8–2.17	0.29, 0.39, 0.48, 0.58, 0.68, 0.78, 0.87, 0.97
8–4.35	0.31, 0.41, 0.51, 0.61, 0.71, 0.82, 0.92
8–6.52	0.26, 0.34, 0.43, 0.52, 0.60, 0.69, 0.77, 0.86, 0.95
10–2.17	0.28, 0.37, 0.46, 0.56, 0.65, 0.74, 0.84, 0.93
10–4.35	0.26, 0.35, 0.44, 0.52, 0.61, 0.70, 0.79, 0.87, 0.96
10–6.52	0.32, 0.43, 0.54, 0.64, 0.75, 0.86, 0.97

## Data Availability

The data used to support the findings of this study are included within the article.

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
