# Peer review of "Experimental Study of Energy Evolution at a Discontinuity in Rock under Cyclic Loading and Unloading"

_materials, 2022, doi:10.3390/ma15165784_

Round 1
Reviewer 1 Report
Introduction
1- Line 3, Authors should write dams and not DAMS
Test equipment
2- Authors should remind readers of the principles of CSS !
Sample preparation
3- In this paper, the authors evaluate the energy dissipation in discontinuities. Where are the discontinuities in your samples?Test Procedure
4- Similarly, what justifies the choice of load rate values of 0.4 kN/s and 0.2kN/s for the tests?
Results and discussion
5- The expressions: load curve, unloading curve, reloading curve are mentioned in the presentation of figure 4 but are not really visible on figure 4;
6- On figure 4, the graduations are blurred and improved.
7- In this work, you have chosen to study only one group of data. Why this choice and which data are useful to make this work complete? We really want the rest of the data to be taken into account.
8- What is the relationship between the parameters of the uniaxial compression test?
9- What is the relationship between the parameters of the biaxial compression test?
10- What is the relationship between the parameters of the graded cyclic shear test?
11- Explain further the process of energy densities.
12- Give the relationships that led to the energy values.
13- The conclusion should be revised as it is only a list of the results obtained.
14- Make the stress level curves explicit.
15- Give the proportions of the correlation between the limit energy storage stress of the sample and the increase of the JRC.
16- Is figure 5 yours? If so, how did you obtain it?
Reviewer 2 Report
Good work, and well written.
Novelty and original contribution of the study could be highlighted more, especially with reference to published work on energy studies for rock under cyclic loading.
Author Response
Thank you very much for your recognition of our work! I wish you success in your work!
Reviewer 3 Report
The manuscript titled: “Experimental study of energy evolution at a discontinuity in rock under cyclic loading and unloading” presents an interesting study that emphasizes on the damage and failure of rock discontinuities from an energy perspective. Specifically, the authors concentrated on the energy evolution of during the process of deformation and failure in order to provide a theoretical basis of the concept. For this reason, samples were made from cement mortar and their Barton’s standard profile was produced. Afterwards, shear, uniaxial compression and cyclic shear tests were conducted so as to determine the shear strength of rock discontinuities, the normal stress value of shear test and the actual loading stress, in respect. By obtaining the results of the aforementioned tests, the energy evolution of structural plane under cyclic loading was depicted by a stress-strain curve. According to the stress-strain curve characteristics of each sample, the elastic energy density and dissipation energy density of rock mass under cyclic loading were calculated using an integral equation. Then, the relationships between (a) the energy density and cycle times under various cyclic stresses, (b) the average energy density and the stress level and (c) the energy ratio and stress under different working conditions, were shown through relevant diagrams. Finally, the energy dissipation ratio was used to characterize the internal damage accumulation of the loaded rock sample and stress level-energy dissipation ratio-deformation diagrams under graded cyclic loading and unloading were constructed.
The presented research corresponds to the aims and scopes of the Journal Materials. The manuscript is quite well written and has the appropriate length. The English language is used fine, but some comments have been made. The authors have used the appropriate citation throughout the manuscript. Even though a completed and well-structured research is provided, some clarifications, changes and additions are recommended; for instance, it is suggested to compare the results of the research with other literature reference, if possible. The presented Tables and Figures are well displayed. The conclusions of the research are supported by the results. Therefore, a Minor Revision is recommended.
Specific comments:
1 In section “Introduction”, when you use the sentence “The research into the development of the theory of rock behavior has been focused on the energy perspective, but the object of study focused on the intact rock, with less research into rock structure;…”, is it possible to use parts of speech that define whether you are referring to your study or to the general work that has been done so far? Please rephrase as suggested.
2 Why do you write the word “DAMS” with capital letters? Is it a mistake? Otherwise, explain why.
3 Can you explain what you mean when you say “the cement mortar samples with 1#,4#, 8#, and 10#...”? Is it possible that you meant “the cement mortar samples maned 1#,4#, 8#, and 10#”?
4 It is suggested to specify how many cement mortar samples were produced and tested. In section “Sample preparation” the code names used, make clear that they were four, in total. However, in section “Uniaxial compression test” you state the following “…uniaxial compression tests were conducted on five complete cement mortar specimens…”. Table 2 reveals that 12 samples were produced. Please make the appropriate changes in the manuscript. For instance, you could say “In total, X cement mortar specimens were produced.” Then, you can explain that 12 were used for the conduction of the shear tests, etc. Further, it is implied that you grouped the samples, when you state “Cement mortar samples with 1#, 4#, 8#, and 10# (JRC = 1, 7, 15, 19)” at every test that you performed. Please, clarify.
5 Some sentences seem to lack conjunction words. For example, the sentence “The occurrence and accumulation of irreversible deformation in the loading and unloading process is the direct cause of the deformation and failure of samples [31], the dissipated energy can indirectly reflect the irreversible plastic deformation in the sample.”, would be expressed in a better way if you have added an “and” as follows: “The occurrence and accumulation of irreversible deformation in the loading and unloading process is the direct cause of the deformation and failure of samples [31] and the dissipated energy can indirectly reflect the irreversible plastic deformation in the sample.” However, you might prefer to use other conjunction word, depending on the meaning you want to express. Please, make the appropriate change. It is, also, recommended that you considered the comment for other sentences throughout the manuscript.
6 It is suggested to rephrase the sentence “The connection and connection of cracks results in the worsening of the…”, as you repeat the word “connection”.
7 Please rephrase the sentence “Instability and failure of specimen occurred after reaching energy storage limit is the root cause of a rapid release of elastic energy, …” as follows “Instability and failure of specimen occurred after reaching energy storage limit, are the root cause of a rapid release of elastic energy, …”, as instability and failure are two causes not one.
8 It is recommended commenting your results with other studies, if possible. Most of the reference that you provided concern the “Introduction” section.
9 Is it possible to connect your conclusions with as you say in the “introduction” section “studying the damage and failure of rock discontinuity from an energy perspective can provide a new idea for preventing and treating rock engineering disasters.”? It is not necessary to be extensive. By explaining this, the originality of your study can be promoted further.
1.
Author Response
Response to Reviewer 3 Comments
Point 1: In section “Introduction”, when you use the sentence “The research into the development of the theory of rock behavior has been focused on the energy perspective, but the object of study focused on the intact rock, with less research into rock structure;…”, is it possible to use parts of speech that define whether you are referring to your study or to the general work that has been done so far? Please rephrase as suggested.
Response 1: Thank you very much for your suggestion. I have revised the “Introduction” section of the paper as you suggested.
Point 2: Why do you write the word “DAMS” with capital letters? Is it a mistake? Otherwise, explain why.
Response 2: Thank you very much for your suggestion. This was a mistake, which I have corrected in the paper.
Point 3: Can you explain what you mean when you say “the cement mortar samples with 1#,4#, 8#, and 10#...”? Is it possible that you meant “the cement mortar samples maned 1#,4#, 8#, and 10#”?
Response 3: Thank you very much for your suggestion. It was my description that caused the ambiguity, and I have revised it in the paper as you suggested.
Point 4: It is suggested to specify how many cement mortar samples were produced and tested. In section “Sample preparation” the code names used, make clear that they were four, in total. However, in section “Uniaxial compression test” you state the following “…uniaxial compression tests were conducted on five complete cement mortar specimens…”. Table 2 reveals that 12 samples were produced. Please make the appropriate changes in the manuscript. For instance, you could say “In total, X cement mortar specimens were produced.” Then, you can explain that 12 were used for the conduction of the shear tests, etc. Further, it is implied that you grouped the samples, when you state “Cement mortar samples with 1#, 4#, 8#, and 10# (JRC = 1, 7, 15, 19)” at every test that you performed. Please, clarify.
Response 4: Thank you very much for your beneficial suggestion. As you suggested, I have added the number and use of the samples in the paper.
Point 5: Some sentences seem to lack conjunction words. For example, the sentence “The occurrence and accumulation of irreversible deformation in the loading and unloading process is the direct cause of the deformation and failure of samples [31], the dissipated energy can indirectly reflect the irreversible plastic deformation in the sample.”, would be expressed in a better way if you have added an “and” as follows: “The occurrence and accumulation of irreversible deformation in the loading and unloading process is the direct cause of the deformation and failure of samples [31] and the dissipated energy can indirectly reflect the irreversible plastic deformation in the sample.” However, you might prefer to use other conjunction word, depending on the meaning you want to express. Please, make the appropriate change. It is, also, recommended that you considered the comment for other sentences throughout the manuscript.
Response 5: Thank you very much for your suggestion. I have added the conjunction words in the corresponding sentences as you suggested.
Point 6: It is suggested to rephrase the sentence “The connection and connection of cracks results in the worsening of the…”, as you repeat the word “connection”.
Response 6: Thank you very much for your suggestion. I have substituted this word in my paper as you suggested.
Point 7: Please rephrase the sentence “Instability and failure of specimen occurred after reaching energy storage limit is the root cause of a rapid release of elastic energy, …” as follows “Instability and failure of specimen occurred after reaching energy storage limit, are the root cause of a rapid release of elastic energy, …”, as instability and failure are two causes not one.
Response 7: Thank you very much for your suggestion. I have substituted this sentence in my paper as you suggested.
Point 8: It is recommended commenting your results with other studies, if possible. Most of the reference that you provided concern the “Introduction” section.
Response 8: Thank you very much for your suggestion. I have made some substitutions to the references in the “Introduction” section as you suggested.
Point 9: Is it possible to connect your conclusions with as you say in the “introduction” section “studying the damage and failure of rock discontinuity from an energy perspective can provide a new idea for preventing and treating rock engineering disasters.”? It is not necessary to be extensive. By explaining this, the originality of your study can be promoted further.
Response 9: Thank you very much for your beneficial suggestion. I have supplemented the conclusion as you suggested, so that it echoes the “Introduction” section.
We really highly appreciate the reviewers’ carefulness, conscientious, and the broad knowledge on the relevant research fields. The reviewers’ warm work provide me with great help in improving the manuscript. We hope that the correction will meet with approval. Once again, thank you very much for your warm work comments and suggestions.
Wish you all the best!
Sincerely yours,
Zhen Wang